

# Mobile evaporite enhances the cycle of physical–chemical erosion in badlands

Ci-Jian Yang[1, 2], Pei-Hao Chen[1], Erica D. Erlanger[2], Jens M. Turowski[2], Sen Xu[2], Tse-Yang Teng[3],

Jiun-Chuan Lin[1], Jr-Chuang Huang[1]

1.Department of Geography, National Taiwan University, No. 1, Sec. 4, Roosevelt Rd., Taipei 10617, Taiwan.

2.German Research Centre for Geosciences (GFZ), Telegrafenberg 14473, Potsdam, Germany.

3.Sustain-vision Consulting Co. Ltd., Taipei 11168, Taiwan.

*Correspondence to*: Ci-Jian Yang (d03228001@ntu.edu.tw)

**Abstract.** Chemical weathering driven by physical erosion is one of the manifestations of natural processes that strongly affect chemical and solid matter budgets at the Earth's surface. However, the influence of extreme climatic erosion on chemical weathering dynamics is poorly understood. Badland landscapes formed in highly erodible, homogeneous substrates have the potential to respond to individual events on scales that are rapid enough for direct observation. Here, we assess the geochemical and grain-size composition of suspended sediment and riverine chemistry measurements collected from two catchments during the 2017 Nesat and Haitang typhoons in southwestern Taiwan. During the typhoons, the sodium adsorption ratio covaried with suspended sediment concentration, which we attributed to sodium-induced deflocculation. Evaporite weathering at peak rainfall is succeeded by peak silicate weather at maximum discharge, which dominates the weathering signal of the event. Overall, our observations suggest that initial weathering of near-surface evaporite enhances the physical erosion of silicate rock during extreme rainfall events.



## 1. Introduction

Chemical weathering induced by physical erosion controls nutrient supply to ecosystems (Milligan and Morel, 2002), reflects dynamic surface processes (e.g., Calmels et. al., 2011; Clift et. al., 2014; Emberson et. al., 2016; Meyer et. al., 2017), and regulates the global carbon cycle and the evolution of Earth's long-term climate (Berner et al., 1983; Ram et al., 1992; Gaillardet et al., 1999). In most landscapes, physical erosion and chemical weathering operate on geological timescales (e.g., Maher et al., 2014). However, studies show that most erosion occurs during stochastic events, such as storms (e.g., Lee et al., 2020; Wang et al., 2021). In particular, typhoons are able to transport large volumes of water and dissolved solids within hours to days, allowing us to observe the interactions between physical erosion and chemical weathering. Nevertheless, observations of the interaction between extreme physical erosion and chemical weathering dynamics are limited (Meyer et. al., 2017). Lack of high-frequency stream water sampling leads to a fundamental difficulty in constraining the dynamic behavior between physical erosion and chemical weathering during a high discharge period (e.g., a typhoon), which could have key implications for the quantification of topographic responses.

Badlands are landscapes characterized by highly erodible and weathered substrates, which are largely devoid of vegetation. The high erodibility of these landscapes provides a unique opportunity to investigate and quantify denudation processes that operate at short timescales (Cheng et al., 2019; Yang et al, 2019, 2021a; 2021b). Soils that contain clays saturated in sodium ions are particularly vulnerable to erosion by water. Sodium ions alter the layer charge of double-layered clay minerals (i.e. smectite) and cause the clays to deflocculate, which refers to the process of breaking up the clay (and ultimately the soil) into finer particles that are more easily washed away by water (e.g., Faulkner et al., 2004; Mitchell et al., 1993; Rengasamy and Olsson, 1991; Rengasamy et al.,1984; Sherard et al., 1976; Kašanin-Grubin et. al., 2018). Additionally, mineral assemblage affects the stability of soil aggregates; for example, small amounts of smectite in kaolinitic materials cause it to be more dispersive and unstable (Levy et al., 1993).




Previous studies in the badlands of SW Taiwan have revealed that dissolving halite and gypsum at
depth migrate to the slope surface and deposit in desiccation cracks during the dry season (Higuchi et
al., 2013, 2015; Nakata and Chigira, 2009). This produces pore water in the near-surface mudstone
with a concentration of $Na^+$ of $1-3$ million $\mu mol/L$ at $1-2$ cm depth (Nakata and Chigira, 2009). We
hypothesize that the dissolving halite and gypsum re-crystallizes near-surface and is deposited in the
mudstone cracks through capillary action during the dry season. Subsequent extreme precipitation
dissolves the evaporite, which enhances erosion by clay dispersity and further exposes more
weatherable materials, forming a positive feedback cycle. Assuming a mudstone substrate that is
primarily comprised of silicate minerals, we expect that the concentration of the evaporite ions should
be consistent with the changes in the sediment concentration and the concentration of silicate ions.

To investigate the relationship between evaporite dissolution and erosion, we use suspended sediment
concentrations (SSC) and stream chemistry data from two catchments in the badlands of SW Taiwan
(Fig. 1), collected with a temporal resolution of 3 hours. We interpret our observations in badlands to
reflect how the excess sodium enhances physical erosion and chemical weathering during a typhoon
event, and the importance of this process for exposing fresh bedrock available for weathering in the
following dry season.

**2. Geological and Meteorological Setting**
In Taiwan's badlands, the annual precipitation exceeds 2 m, and 90% of the rainfall is concentrated in
the rainy season. The rainy season lasts from May to October and reaches its peak in August, with over
400 mm of precipitation within a single month. In contrast, less than 40 mm of average monthly rainfall
is measured from November to April. We collected river water samples from two sites downstream of
the studied badland areas. The first site, Nanxiong Bridge (NX), is located at the midstream of the
Erren River and has a drainage area of 175 km². This area includes badlands covering an area of 4.37



km$^2$, which accounts for 2.49% of the total catchment area (Fig.1). The Erren River catchment is
predominantly underlain by Plio-Pleistocene mudstones, which are several kilometers thick, and
mainly feature illite (30.54%) and chlorite (28.70%) minerals (Tsai, 1984a). During the dry seasons,
the pore water chemistry in the near-surface mudstones is mainly composed of $Na^+$, $Cl^-$, $Ca^{2+}$ and $SO_4^{2-}$
(Nakata and Chigira, 2009).

The gauging station at Nanxiong Bridge (NX) provides hourly discharge data for calculating sediment
and solute fluxes. The annual average discharge of Nanxiong Bridge station is 10.2 m$^3$/s, and the
typhoon season accounts for 84% of the total discharge. The meteorological station at Gutingkeng
(GTK) is located 5.5 km from Nanxiong Bridge and provides hourly precipitation data. Our second
sampling site is Guting (GT) Bridge, with an upstream drainage area of 79 km$^2$ and a badlands area of
1.87 km$^2$, corresponding to 2.37% of the total area. Guting Bridge is located adjacent to a badlands
conservation area, so the riverine water chemistry reflects the weathering products derived from the
adjacent hillslopes. Due to a lack of stream discharge observations at Guting Bridge, we use hourly
precipitation data at GTK, which is less than 1 km from the sampling site, to quantify the impact of
the typhoon events.



Earth **Surface**
**Dynamics** Open Access
Discussions

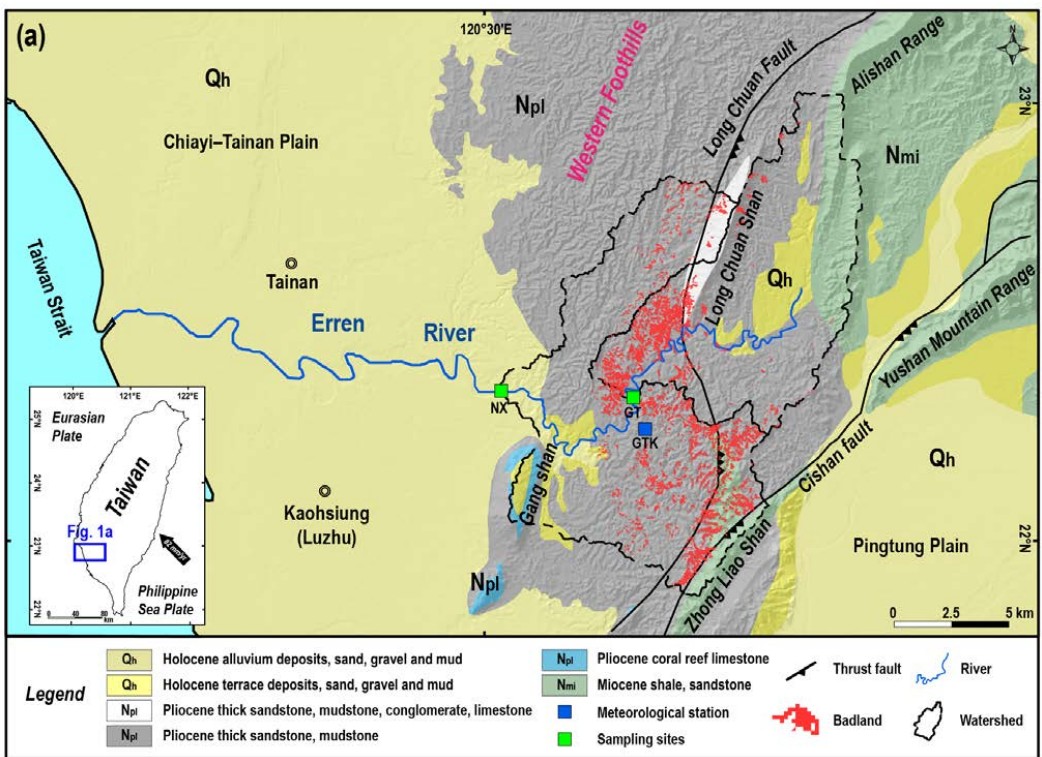


**Figure 1.** Location of sampling sites and geology of the study area. (a) The geological map of the study area (Source: Central Geological Survey, 2013). The green squares are sampling sites; hourly stream discharge data were obtained from the Nanxiong Bridge (NX) hydrometric station (Water Resources Agency). The blue square is the meteorological station, which provides hourly precipitation data (Central Weather Bureau, https://dbar.pccu.edu.tw/).

**3. Methods and Materials**

3.1 Water Sampling

We collected 42 stream samples from the two sampling sites for the typhoon period of July 2017. During sample collection, two 1000 ml PE bottles were dropped 1 to 2 meters below the water surface of the river simultaneously. Suspended sediment concentration (SSC) was subsequently calculated from the water collected in one of the PE bottles, and riverine chemistry was determined from water collected in the other bottle. Samples were filtered *in situ,* and the filtrate was preserved in the refrigerator for laboratory analysis. Additionally, 31 samples were collected from September 2014 to



December 2016 in the second half of every month at Nanxiong Bridge for non-typhoon periods, using
the same sampling procedure.

**3.2 Dissolved load and sediment chemistry analysis**
For the riverine dissolved load, we measured major dissolved anions ($Cl^-$, $SO_4^{2-}$, $NO_2^-$, $NO_3^-$, $F^-$) on an
Ion chromatography (IC, Metrohm Basic-883 plus), and we measured major dissolved cations ($Na^+$,
$K^+$, $Mg^{2+}$, $Sr^{2+}$, $Ba^{2+}$, $Si^{4+}$) on an ICP-OES (PerkinElmer, Optima 2100DV). We measured bulk
sediment chemistry from two samples of suspended sediment collected from Guting Bridge at low
flow before the typhoon event (2.26 $m^3$/s) and at the peak of runoff (724.32 $m^3$/s). About 0.7 g of dried
sediment sample was combusted in the muffle furnace at 650°C for 2 hours and then weighed to obtain
the loss on ignition (LOI). Afterwards, an aliquot of ~100 mg from the residue was digested with a
mixture of concentrated HF and aqua regia. After digestion and drying, the sample was dissolved in
0.3 N $HNO_3$ for elemental determination. Major elemental concentrations of sediment samples were
obtained by ICP-OES (Varian 720-ES) at the GFZ German Research Centre for Geosciences.

**3.3 Grain size of suspended load**
Before measuring grain size, we removed the non-clastic deposition, i.e., sea salt, organic matter, and
carbonate. To remove sea salt, ~1 g of dried sediment sample was added to 15 ml of distilled water,
placed in a shaker, and shaken at a speed of 4000 rpm for 5 minutes. The centrifuged supernatant was
then poured out and these steps were repeated 3 times. To remove organic matter, 10 ml of a 15%
$H_2O_2$ solution was added to the sediment and placed in an ultrasonic oscillator for 24 hours. After
adding a second 10 ml of $H_2O_2$ (15%) to confirm the completion of the reaction, the mixture was
centrifuged and the supernatant containing the organic matter was removed. The sediment was then
washed by adding 30 ml of distilled water, and the supernatant was again removed after centrifugation.
This washing step was repeated 3 times to remove residual $H_2O_2$ in the centrifuge tube. To remove the



carbonates, we added 10 ml of 10% HCl solution to the centrifuge tube and allowed the acid to react
with the sediments for 24 hours. An additional 10 ml of HCl was then added to confirm the
completeness of the reaction. The sample was then centrifuged, and the supernatant was decanted to
remove the carbonates. The sample was then rinsed with 30 ml of distilled water, centrifuged, and
decanted. This step was performed 3 times to remove any residual HCl.

To disperse sediment agglomeration, we added 10 ml of 1% $Na(PO_3)_6$ solution to the sediment and let
the sample react for more than half a day. The grain size of the sediment samples was obtained by
Laser Diffraction Particle Size Analyzer LA950 at the GFZ German Research Centre for Geosciences.
By using LA950, we measured grains in the size range of between 100 nm to about 3 cm.

**3.4 Calculation of the enriched ratio and sodium adsorption ratio (SAR)**
In order to classify the supply of different ion sources during the typhoon event, we used the enriched
ratio of concentration as a reference. The enriched ratio is the ion concentration at a certain time
divided by the ion concentration at the first observation. A value greater than 1 represents a point in
time when the sample is more concentrated relative to the first observation, whereas a value smaller
than 1 represents a point in time when the sample is more diluted relative to the first observation.

Dissolved calcium and magnesium can stabilize soil aggregates and therefore facilitate water
permeability (Nadler et al., 1996). By contrast, excess sodium can disperse soil particles through
deflocculation, thereby reducing water permeability (Hanson et al., 1999). The potential for material
dispersion in badlands is generally determined by measuring the presence and behavior of sodium and
is quantified by the sodium absorption ratio (SAR), (1):
$$SAR = \frac{Na^+}{\sqrt{\left(\frac{Ca^{2+}+Mg^{2+}}{2}\right)}} \qquad\qquad (1)$$



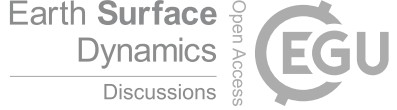

When SAR is greater than 13, the excess sodium causes soil particles to repel each other, preventing
the formation of soil aggregates (Seelig, 2000; Horneck et al., 2007).

**3.5 Calculation of TDS and chemical weathering rate**
Riverine TDS is widely used to estimate chemical weathering rates of river catchments (e.g. Gaillardet
et al. 1999). In this study, riverine TDS (in units of µmol/L) is expressed as:
$TDS = TDS_{rain} + TDS_{evaporite} + TDS_{sil} + TDS_{carb}$     (2)
where the contributions from precipitation ($TDS_{rain}$), evaporite ($TDS_{evaporite}$), silicate weathering
($TDS_{sil}$) and carbonate weathering ($TDS_{carb}$) are considered. We calculated the proportions of ion
contributions with the MEANDIR inversion model (Kemeny and Torres, 2021), a MATLAB script for
inverting fractional contributions of end-members, and for constraining the chemical compositions of
those end-members. To exclude the input of precipitation ($TDS_{rain}$) from riverine TDS, we used local
rainwater $Cl^-$ concentrations with an average value of 68 µmol/L (Lu, 2014), and also the ratios of $Cl^-$
and $SO_4^{2-}$, $Na^+$, $K^+$, $Mg^{2+}$, $Ca^{2+}$ in rainfall based on the rainfall chemistry from 2007 to 2013 ($SO_4^{2-}$/
$Cl^- = 0.35$, $Na^+/Cl^- = 0.90$, $K^+/Cl^- = 0.09$, $Mg^{2+}/Cl^- = 0.18$, $Ca^{2+}/Cl^- = 0.35$) (Lu, 2014). Thus, we
estimated the annual deposition of those cations using equation (3):
$[X]_{norain} = [X]_{river} - [X]_{rain}$     (3)
$TDS_{rain} = \sum [X]_{rain}$     (4)

Here $[X]_{norain}$ reflects the remaining concentration of ion X after the removal of atmospheric

inputs; $[X]_{river}$ is the concentration of ion X in river water, and $[X]_{rain}$ is the concentration of ion X from
atmospheric deposition. In the second step, we corrected for evaporite inputs ($TDS_{evaporite}$) using the
following equation:
$[X]_{NSS} = [X]_{norain} - [X]_{evap} = [X]_{norain} - \left( [Cl]_{norain} \times (\frac{X}{Cl})_{evap} \right)$     (5)
$TDS_{evaporite} = \sum [X]_{evap}$     (6)



where $[X]_{NSS}$ is the concentration of ion X after the removal of ions attributed to evaporites, $[X]_{evap}$.
$[X/Cl]_{evap}$ is the ratio of ion X and Cl by using the end-member molar ratios of evaporite ($SO_4^{2-}/Cl^- =$
0.4, $Na^+/Cl^- = 1$, $Mg^{2+}/Cl^- = 0.10$, $Ca^{2+}/Cl^- = 0.5$, Burke et al., 2018; $K^+/Cl^- = 0.026$, Chao et al., 2013).
Then, after the correction for evaporite, the chemical weathering budget can be divided into
contributions by silicate ($TDS_{sil}$) and carbonate weathering ($TDS_{carb}$), expressed as:
$$TDS_{sil} = [Na]_{sil} + [K]_{sil} + [Mg]_{sil} + [Ca]_{sil} + [SiO_2]_{sil} \tag{7}$$
$$TDS_{carb} = [Mg]_{carb} + [Ca]_{carb} + [HCO_3]_{carb} \tag{8}$$
$$[HCO_3]_{carb} = \frac{1}{2}([Mg]_{crab} + [Ca]_{crab}) \tag{9}$$
where $[Na]_{sil}$ and $[K]_{sil}$ are riverine $[Na]_{NSS}$ and $[K]_{NSS}$ concentration, respectively. We used
endmember values for silicate- and carbonate-dominated rocks reported by Gaillardet et al. (1999),
which gave ratios of Ca/Na =0.35 and Mg/Na =0.24 for silicates, and Ca/Na = 50 and Mg/Na =10 for
carbonates.

**4. Results**
**4.1 Geochemistry of river water and suspended sediment**
In 2017, the Nesat and Haitang typhoons brought 579 mm of rainfall over three days, with a maximum
intensity of 74 mm/hr. The discharge at Nanxiong Bridge demonstrated that the climatic co-response
has two pulses (Fig. 2). Since the time interval between the two typhoons was less than 6 hours, we
define the two typhoons as one typhoon event and distinguish between a first and second discharge
pulse. We quantify time relative to the onset of the typhoon (0 hr). The first pulse occurred from 8.5 to
32.5 hr, with a mean water discharge of 66.2 m³/s. The second pulse that occurred from 32.5 to 62.5
hr had a 5.5 times higher mean discharge of 369.2 m³/s. The maximum discharge (753.2 m³/s) was
observed during the second pulse at 44.5 hr (July 31th, 2017, at 6:00 a.m.) (Fig. 2).



At Nanxiong Bridge, SSC has a statistically significant positive correlation with SAR ($\rho = 0.51$, $p <$
0.05). SSC has two peaks during the both pulses, but SAR only shows a peak during the first pulse.
During the first pulse, SSC ranged from 10 to 33757 mg/L and SAR increased from 8.2 and to 17.7.
During the second pulse, SSC increased from 5445 to 16900 mg/L and SAR is steadily about 7.3.
$D_{50}$ ranged from 3.9 to 8.2 µm, with an average value of 5.6 µm during the second pulse, and exhibited
a positive correlation with discharge ($\rho = 0.40$). At Guting Bridge, SSC has a statistically significant
positive correlation with SAR ($\rho = 0.69$, $p < 0.05$) during the survey. SSC ranged from 164 to 19538
mg/L before the first pulse and ranged from 2857 to 35920 mg/L during the second pulse, while SAR
showed a mean of 8.2 and two peaks with a value over 20 during both pulses. $D_{50}$ ranged from 3.6 to
8.8 µm, with an average value of 5.3 µm during the second pulse, (Fig. 2). In terms of sediment
chemistry at Guting Bridge, major elements of the two selected sediment samples show that calcium
and sodium accounted for about 10% of the mass loss between the typhoon event (5.5 hr of duration)
and peak of discharge (41.5 hr of duration) (Table. S4).
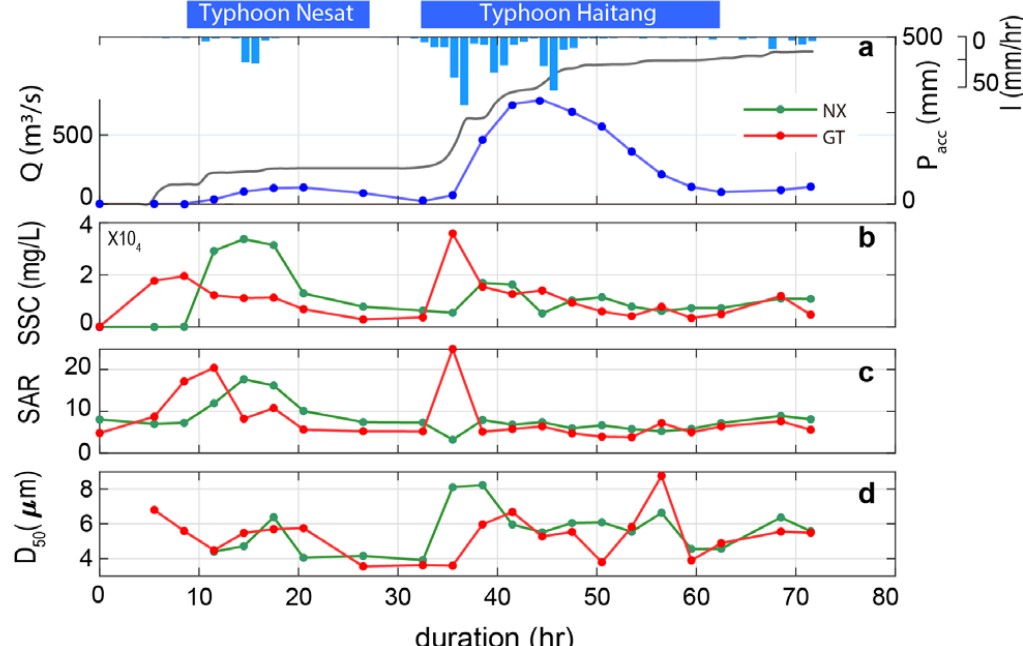

**Figure 2.** Timeseries SSC, SAR and median grain size of suspended sediment ($D_{50}$) at two sampling
sites. The blue line denotes hourly discharge (Q) at Nanxiong Bridge, and the blue bar denotes hourly





precipitation (I) at Gutingkeng station. The gray line denotes precipitation accumulation ($P_{acc}$), the
green line denotes the Nanxiong Bridge (NX) dataset, and the red line denotes the Guting Bridge (GT)
dataset.

The fractional proportions of TDS at Nanxiong Bridge during baseflow show that precipitation,
evaporites, silicates, and carbonates contribute 3.0±1.1%, 28.7±14.6%, 26.9±6.5%, and 41.4±13.2%,
respectively (Fig. 3a). During the typhoon event, the proportion of TDS at Nanxiong Bridge attributed
to $TDS_{rain}$ is 6.3±2.4%. $TDS_{evaporite}$ contributes 32.4±13.6% and increases from 27.4% to 61.1% at the
incipient first pulse. $TDS_{sil}$ contributes 39.5±15.2%, which is 12.6% higher than the non-typhoon
period. $TDS_{carb}$ contributes 21.8±11.5 % (Fig. 3b), which is 19.6% lower than the non-typhoon period.
The fractional proportions of TDS at the Guting Bridge show that 6.5±2.1% of TDS is contributed by
$TDS_{rain}$. $TDS_{evaporite}$ contributes 24.8±16.2% and increases from 13.6% to 61.6% at the incipient second
pulse, when the SSC and SAR peak simultaneously. $TDS_{sil}$ and $TDS_{carb}$ contribute 39.5±15.2% and
27.5±16.7%, respectively (Fig. 3c).

Enriched ratios less than 1 indicate dilution, and values greater than 1 indicate concentration. Since we
set the ion concentration of rainfall to be constant during the typhoon event, the enriched ratio of
precipitation is constant throughout the observation period. At Nanxiong Bridge, the evaporites
enriched ratio increases from 0.4 to 1.7 between the two pulses and decreases to 0.1 at the discharge
peak. The silicates enriched ratio increases from 1 to 1.5 before the first pulse and decreases to 0.1 at
the peak of discharge, then returns to 1 before the observation ends. The concentration attributed to
carbonates is always diluted. The evaporites and carbonates enriched ratio have a statistically
significant negative correlation with discharge (evaporites: $\rho$ = -0.67, carbonates: -0.60, p<0.05) and
the silicate enriched ratio has a negative correlation with discharge ($\rho$ = -0.32), indicating dilution by
typhoon rainfall (Fig. 3d). At Guting Bridge, the evaporites enriched ratio has two peaks during the
two pulses with a value of 5.2 at the first peak, a value of 4.7 at the second peak. After the event, the
value returns to about 1.2. Notably, the evaporites enriched ratios during the both pulses are similar,





but the peak discharge of the second pulse is 5.5 times higher than that of the first pulse. The silicate
enriched ratio has an analogous pattern with the evaporites enriched ratio, but the enriched ratio is
smaller. Similar to Nanxiong Bridge, the carbonates enriched ratio is always diluted at Guting Bridge
(Fig. 3e). The evaporite and silicate enriched ratio shows a statistically significant positive correlation
($\rho = 0.96$, p<0.05), and the evaporite and silicate enriched ratios have a statistically significant positive
correlation with SAR ($\rho = 0.86$, $\rho = 0.84$, p<0.05).

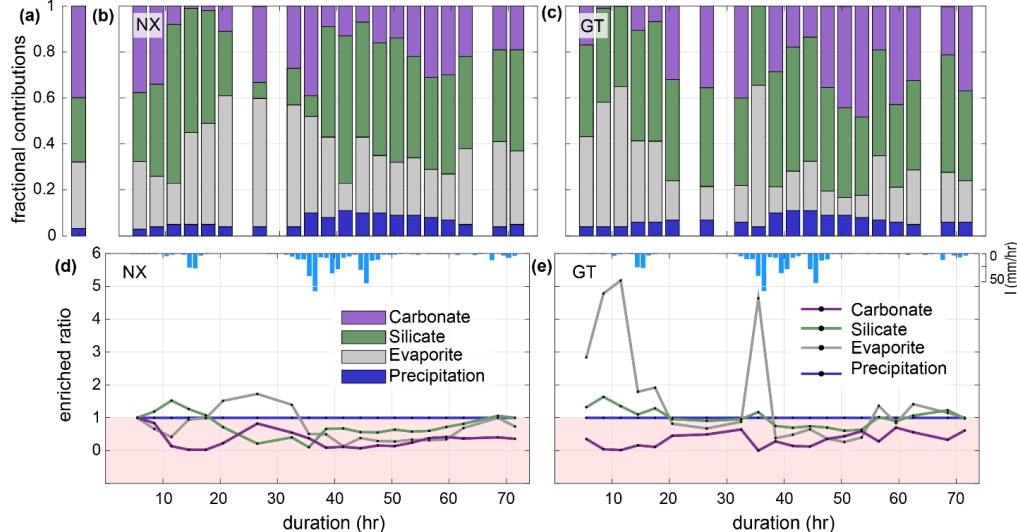


**Figure 3.** Timeseries illustrating TDS sources during the typhoon event at the two sampling sites.
Fig.3a shows the average proportion of TDS for the non-typhoon period from September 2014 to
December 2016 at Nanxiong Bridge; Fig.3b-c denotes the endmember contributions to TDS at
Nanxiong Bridge dataset and Guting Bridge dataset from the typhoon period; the purple bar denotes
$TDS_{carb}$ (Eq. 8); green denotes $TDS_{sil}$ (Eq. 7); the gray bar denotes $TDS_{evaporite}$ (Eq. 6); the blue bar
denotes $TDS_{rain}$ (Eq. 4). Fig.3d-e denotes the enriched ratio of ion concentrations by TDS sources from
the Nanxiong Bridge dataset and Guting Bridge dataset during the typhoon period. The purple line
denotes $TDS_{carb}$, the green line denotes $TDS_{sil}$, the gray line denotes $TDS_{evaporite}$, the blue line denotes
the $TDS_{rain}$, and blue bar denotes hourly precipitation (I) at GTK station.



## 4.2 Evaporite dissolution over time

We calculated the enriched ratios of ions (i.e., $Na^+$, $Cl^-$, $Ca^{2+}$ and $SO_4^{2-}$) that are sourced from evaporites (i.e., halite (NaCl) and gypsum ($CaSO_4$)). The variability in the concentrations of each of these ions reflects the overall trends in TDS (Fig. 3d-e & Fig. 4).

At Nanxiong Bridge, all evaporite and carbonate ions have a statistically significant negative correlation with discharge. The enriched ratios in evaporite $Na^+$, $Cl^-$ and $SO_4^{2-}$ have the same trend (Fig. 4), which show an initial decrease during the first pulse, followed by an increase to 2 between the two pulses, and a final decrease during the second pulse. Evaporite $Ca^{2+}$ shows a similar trend with evaporite $Na^+$, $Cl^-$ and $SO_4^{2-}$, but the values are below 1. The enriched ratios of silicate $Na^+$, $Ca^{2+}$, and $SO_4^{2-}$ show an increase during the first pulse and a decrease to less than 1 before the rainfall peak, followed by an increase from about 0.06 to 1.11 at the end of observation. At Guting Bridge, all evaporite ions have a statistically significant positive correlation with the corresponding silicate ions ($Na^+$=0.98; $Ca^+$=0.81; $SO_4^{2-}$=0.98, $p<0.05$). Evaporite $Na^+$, $Cl^-$, and $SO_4^{2-}$ each have two peaks that occur prior to the maximum rainfall and reflect a factor of 5 increase in the enriched ratio. Compared with Nanxiong Bridge (downstream), the enriched ratio in evaporite $Ca^{2+}$ at Guting Bridge concentrates at the onset of the first pulse and after peak discharge. Additionally, the enriched ratios of carbonate at Guting Bridge are similar to Nanxiong Bridge, and are always below 1.

Earth **Surface**
**Dynamics**
Discussions

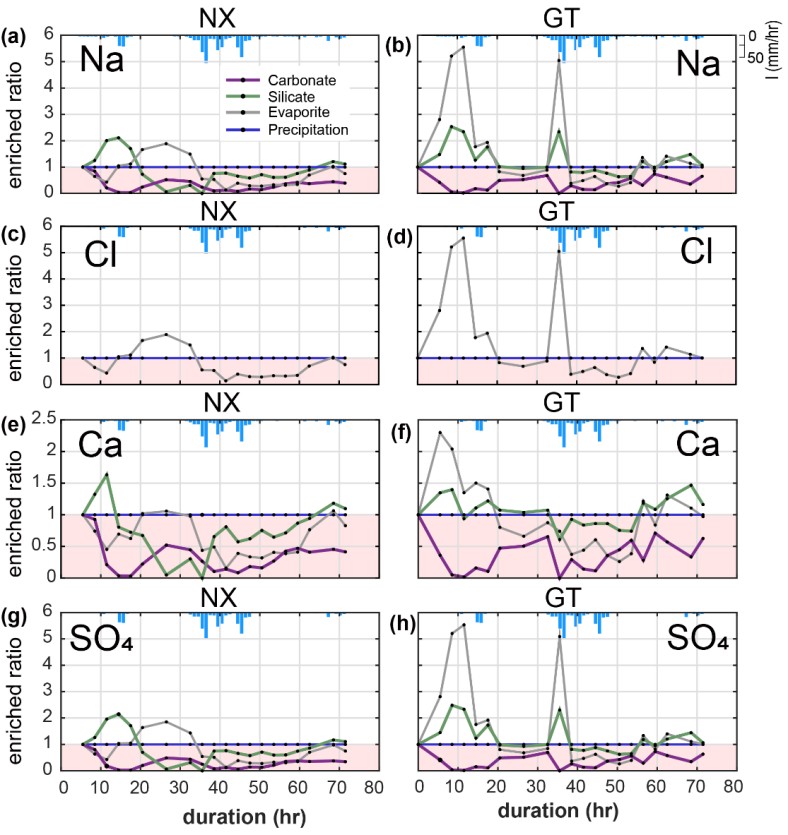

**Figure 4.** Time-series patterns in enriched ratio at two sampling sites. NX denotes the Nanxiong Bridge dataset and GT denotes Guting Bridge dataset. The pink area indicates enriched values below 1. Blue bar denotes hourly precipitation (I) at Gutingkeng station.

Gaillardet et al. (1999) documented that dissolved ions ratios of Ca/Sr and Na/Sr are distinct for carbonates (low Na/Sr, high Ca/Na) versus silicates or evaporites (high Na/Sr, low Ca/Na). We use these ratios to elucidate potential mixing between carbonates and silicates/evaporites (Fig. 5). At Nanxiong Bridge, non-typhoon ratios of Na/(1000*Sr) and Ca/(1000*Sr) are 0.23–0.68 and 0.19–0.35, respectively (Table S4). These values increase markedly during the typhoon events, with enriched ratios of Na+ exceeding 5 at T = 11.5 and 35.5 hr. The high concentration of $Na^+$, $Cl^-$ and $SO_4^{2-}$ (as illustrated in the enriched ratio) indicate that there is enhanced dissolution of evaporites at the onset of the typhoon event, especially at Guting Bridge. Subsequently, the concentration of $Na^+$ decreased with



sustained rainfall. Then, the ratios approach the silicates/carbonates weathering (high Na/Sr, high
Ca/Sr ratios ) after the peak discharge.

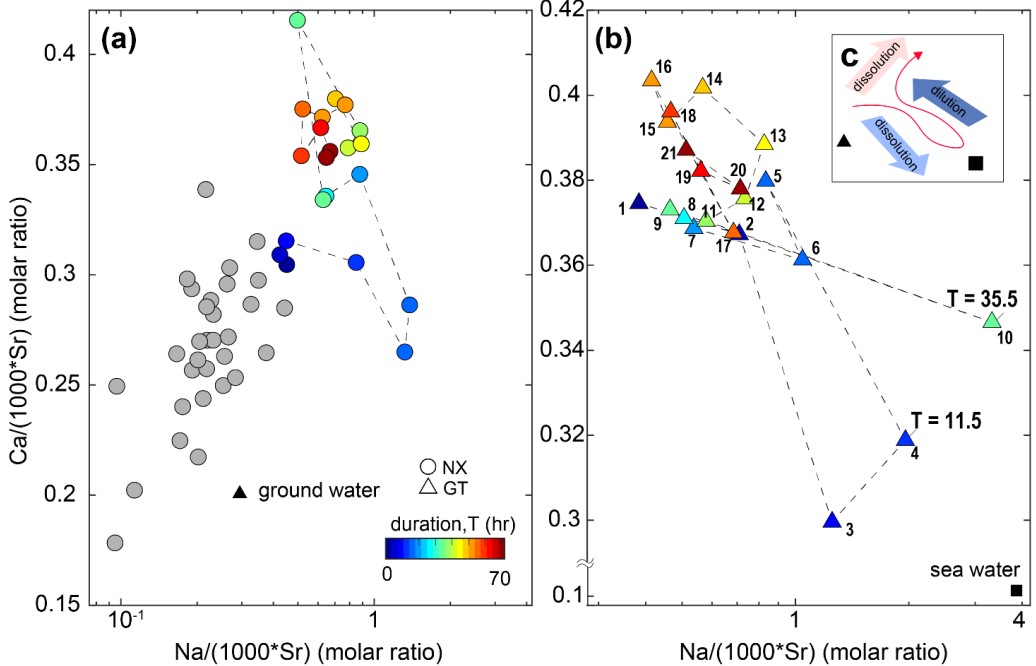


**Figure 5.** Molar ratio mixing diagrams of Erren River waters for (a-b) Na/(1000*Sr) versus
Ca/(1000*Sr), circles denote dataset at Nanxiong Bridge, and triangles denote dataset at Guting Bridge.
Colorbar denotes survey duration. Gray circles denote the dataset at Nanxiong Bridge during baseflow
conditions from 2014 to 2016. The black triangle illustrates the groundwater endmember (Chao et al.,
2011); the black square illustrates the seawater endmember. Numbers in the triangle represent the time
sequence, 1 represents the start point, and 21 represents the end point. (c) Illustration of dynamic
weathering. The red line indicates the direction of change with time. The light blue arrow denotes
dissolution of evaporite, the dark blue arrow denotes dilution from rainfall, and the red arrow denotes
dissolution of suspended sediment.

**5. Discussion**
**5.1 Relationships between dissolved evaporite and river water chemistry**
Before the survey, the monthly rainfall of the study area was 72.5 mm, which is only 18% of the
average monthly rainfall, implying that it provides a relatively dry environment for accumulating



311 evaporites on the slope surface. Under maximum rainfall intensity, $Na^+$, $Cl^-$ and $SO_4^{2+}$ at Guting Bridge

312 show markedly increased concentrations at the onset of the typhoon, peaks in enriched ratios that

313 exceed 5 (Fig. 4), and the greatest contribution of dissolved ions from evaporites (Fig. 3). In addition,

314 the sodium absorptions ratio (SAR) has a statistically significant positive correlation with $TDS_{evaporite}$

315 ($\rho = 0.86$, $p<0.05$) at Guteng Bridge (upstream). During the typhoon event, the SAR increases from

316 4.8 to 24.9 and exceeds the threshold value of 13 at the incipient first pulse and at the incipient second

317 pulse. This pattern indicates that excess sodium is effective at inducing material dispersion and thus,

318 contributing to a higher suspended sediment load (Fig. 2).

320 These observations and results suggest that rainwater in the typhoon event rapidly dissolves the

321 evaporites on the slope surface, which produces high measured concentrations of $Na^+$, $Cl^-$, and $SO_4^{2+}$

322 during the time of peak precipitation (30-40 hr of duration). Furthermore, the dissolution of the near-

323 surface evaporite deposits should be most heavily influenced by runoff from the hillslopes, so we

324 expect that excess sodium and enhanced erosion will be most significant on the hillslopes.

326 At Nanxiong Bridge, we observe a 10-hour delay in the peak enriched ratio relative to the SAR (Fig.

327 3d) and overall lower enriched ratios relative to Guteng Bridge (Fig. 3d-e). We suggest that dilution

328 and the transport distance from the badlands is responsible for this. The two catchments have a similar

329 areal extent of badlands within the total catchment area, which is about 2.49% at Nanxiong Bridge

330 catchment and 2.37 % in Guting Bridge catchment. Badlands contribute considerable evaporite solutes

331 (Chou, 2008), but the higher downstream drainage area will result in dilution of the solutes without

332 additional inputs. Additionally, Nakata and Chigira (2009) have observed that salt dissolution induces

333 an increase in electrical conductivity during intermittent rainfall events and decreases gradually after

334 rainfall events when evaporation and salt precipitate. Therefore, re-crystalization during transportation

335 is to be expected.




### 5.2 From evaporite dissolution to silicate weathering

Our observations show that the water chemistry of the typhoon event is dominated by silicate
weathering at 16.8 ton/km$^2$/day, contributing 16.6% to the annual silicate weathering flux (Table S3).
Additionally, we observed a change in dominant chemical weathering mechanism during the typhoon
event. We rule out significant contributions from groundwater and deep seawater after peak discharge,
since ratios shift to higher Na/Sr, and Ca/Sr ratios relative to the non-typhoon ratio (Fig. 5a–b), and
the Ca/Sr ratio of mud volcanoes in the study site is one order of magnitude less than river water (Chao
et al., 2011). Carbonate weathering is the primary contributor of Ca$^{2+}$ for most of the world's large
rivers (Gaillardet et al. 1999), but the increased Na$^+$ and consistently enriched ratio of carbonate Ca$^{2+}$
does not make this a likely contributor to the Erren River. We thus suggest that the primary contributor
to weathering is from enhanced silicate dissolution. This interpretation is supported by the temporal
evolution of the enriched ratio of silicate Ca$^{2+}$, which gradually increases after the discharge peak, to
approach a value of about 1 at the end of survey (Fig. 4e&f). As such, in the waning of the event,
excess Ca$^{2+}$ originates from a silicate source. Therefore, we suggest that the ratios shift to higher Na/Sr,
Ca/Sr ratios is due to silicate weathering. We also observe that the masses of Na and Ca are reduced
by 10.6% and 9.9%, respectively, in the suspended sediment during the course of the typhoon event
(Table S6).

Given that the sediment transported in the channel is supplied by physical erosion, we suggest that
physical erosion in our study site enhances silicate chemical weathering, which is consistent with
previous studies (Chung, 2002; Chou, 2008). Thus, we associate the change in weathering regime
during the course of the typhoon with abrasive erosion of silicate sediments in the channel. Mudstone
is mainly composed of silicate minerals (e.g., illite and chlorite minerals) (Tsai, 1984a), and few
swelling clay minerals (e.g., montmorillonite), which provide an abundant silicate pool. We suggest



that high suspended sediment concentrations combined with high energy flow during the typhoon,
caused increased silicate input from the weathered silicates in the suspended sediment. This trend can
explain about 10% of the reduced mass and it has also been observed on typhoon-driven silicate
chemical weathering from silicate minerals at surface (Meyer et. al., 2017). Importantly, the silicate
weathering flux that we calculate in this study is comparable to the global annual flux of rivers
(Gaillardet et al. 1999), suggesting that individual stochastic events may have global relevance.

**5.3 Typhoon-controlled cycles of physical and chemical erosion**
Evaporites, including halite ($NaCl$) and gypsum ($CaSO_4$), are found in few sedimentary environments,
and they are often excluded from global chemical weathering cycles (Gaillardet et al., 1999).
Compared to silicate rocks, the relation between evaporites weathering and physical erosion has rarely
been discussed. Through the interactions among riverine chemistry, suspended sediment properties,
and previous soil water chemistry studies, we suggest a positive feedback cycle of physical-chemical
erosion driven by mobile dissolved evaporite (Fig. 5). The feedback cycle includes three steps. (1)
precipitation and deposition of evaporite during the dry season in near-surface mudstone desiccation
cracks through capillary transport (Higuchi et al., 2013, 2015; Nakata and Chigira, 2009). In the dry
season, exposed bedrock with low water content develops desiccation cracks (Allen, 1982; Goehring
et al., 2010; Kindle, 1917; Seghir and Arscott, 2015; Xiaa and Hutchinson, 2000), providing space for
the re-precipitation of evaporite minerals. Using evidence from core samples in mudstone bedrock at
the study site, the depth of the crack of about 20 cm can be regarded as the thickness of the weathering
layer. Higuchi et al. (2013) suggested that the weathering layer in the top 10 cm of mudstone can easily
be eroded by intense rainfall. Erosion exposes fresh bedrock, which would dry in the following dry
season and further produce weatherable material.

(2) Rainfall dissolves the evaporites, producing sodic water that increases physical erosion during
typhoon events. The resulting dissolved sodium causes higher hillslope erosion by deflocculation,

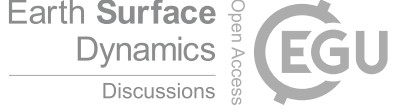



leading to increased suspended sediment in the channels. In the study site, hillslope erosion rate is
about 9-30 cm/year (Higuchi et al., 2013; Yang et al., 2021a). At Nanxiong Bridge, the denudation rate
approaches about 142,857 ton/km$^2$/yr, measured from river suspended load (Dadson et. al., 2003), and
the chemical weathering flux is 124-235 ton/km$^2$/yr (Chou, 2008; this study). The high hillslope
erosion rate ensures a steady supply of freshly exposed bedrock, allowing for high chemical weathering
rates.

(3) Physical erosion enhances silicate weathering and bedrock exposure on hillslopes. Clay minerals
in mudstone deposits are abraded from the abundantly available sediment and provide material for
silicate weathering in streams. Ultimately, with frequent typhoon events and high temperatures in the
study area, this dynamic cycle could repeat several times a year.

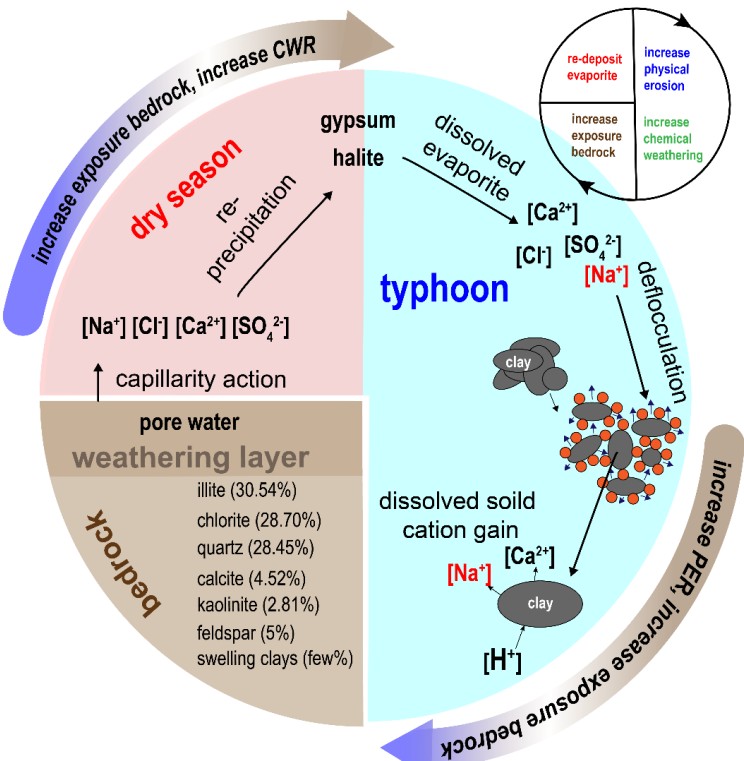


**Figure. 6.** Cycle of feedback between physical erosion rate (PER) and chemical weathering rate (CWR)



in badlands catchment. Red blocks represent dry season conditions. Blue region represents typhoon
conditions. Brown region represents the bedrock and indicates the type and proportion of minerals of
mudstone (Tsai, 1984b).

**6. Conclusion**
We presented major element compositions of stream water from two sites in the Erren River catchment
at three-hour intervals during a three-day typhoon event in 2017. At the Guteng Station (upstream),
$TDS_{evaporite}$ is covariant with $TDS_{sil}$, the sodium adsorption ratio, and the suspended sediment
concentration, which can be assigned to dissolved evaporite (e.g., halite and gypsum). The excess
sodium in the evaporite deposits causes material dispersion through deflocculation, which enhances
the suspended sediment flux. Our observations show that the water chemistry of the typhoon event is
dominated by silicate weathering at 16.8 ton/km$^2$/day, in contrast with baseflow (non-typhoon)
conditions that are dominated by carbonate weathering. Moreover, during the course of the typhoon,
we observed a shift from predominantly evaporite weathering during peak precipitation to silicate
weathering at peak discharge.

Combining the observation of riverine chemistry, suspended sediment properties, and previous soil
water chemistry studies, we propose a feedback cycle between physical erosion and chemical
weathering in badlands topography, illustrating that precipitation of evaporites during the dry season
produces sodic water during typhoon events and preferentially triggers higher local erosion. The
enhanced hillslope erosion and abrasive effects of clay in a high discharge stream enhance bedrock
exposure on hillslopes and silicate weathering, respectively. Newly exposed bedrock then produces
more weathered material. Although measurements of bedrock mineral chemistry and Sr isotope are
still needed for confirming sources of excess sodium and calcium (Fig. 5), we suggest that the
conceptual model could provide an insight into landscape change of badlands. The results from our
study suggest that high erosion rates in mudstone badlands of the Erren River catchment is due to both
weakened lithology and to the interaction between evaporites and hillslope erosion.





*Data availability.* Relevant data supporting the findings of the study are available in the Supplementary

Information, or from the corresponding author upon request. Source data are provided with this paper.

*Author contributions.* C.-J.Y. designed the study and conducted field surveys, data analysis, and

modelling. P.-H. C. conducted data analysis. C.-J.Y., E. D. E. and J.M.T. wrote the paper with input

of all authors. S. X. conducted modelling. T. Y. T. provided the verified data. J.-C.L. and J.-C. Huang

contributed to the scientific discussion, interpretation, and paper preparation.

*Competing interests.* The authors declare that they have no competing interests.

*Acknowledgements.* This is study was supported by grants from National Science and Technology

Council, Taiwan to Ci-Jian Yang (MOST 110-2917-I-564-009-).

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
