# Peer review of "Mobile evaporite enhances the cycle of physical-chemical"

_Earth Surface Dynamics, 2022_

## Author Comment (AC1)

**Response Letter**

We thank the reviewers for their insightful and constructive comments and corrections, which helped us to greatly improve the manuscript. We address their concerns point by point, and highlight implemented changes in the manuscript.

Referee #1

Lines 56-58 – This sentence is confusing. Do you mean that if you assume the mudstone is primarily comprised of silicate minerals then you expect the silicate weathering rate to co-vary with Na concentrations from evaporite weathering? Or do you mean that you expect concentrations of SiO2 and SSC to covary with Na?

Response: Badland landscape is rich in silicate minerals derived from the mudstone, as well as layers of evaporite minerals. The dissolution of evaporite liberates Na ions, which enhances hillslope erosion and causes high concentrations of suspended load. High proportion of silicate minerals can contribute to dissolved ions by at least two approaches, i.e., silicate weathering from silicate minerals at the surface during dry season (e.g., Calmels et al., 2011.https://doi.org/10.1016/j.epsl.2010.12.032) and mass loss of silicate minerals in the stream. Assuming an enriched ratio above 1, we expect that Na concentrations from evaporite weathering should co-vary with SSC and the silicate weathering rate.

We revised the sentence as follows:

"*Subsequent precipitation dissolves the evaporite, and the dissolved $Na^+$ enhances erosion by clay dispersity and exposes more weatherable materials, forming a positive feedback cycle. Assuming a mudstone substrate that is primarily comprised of silicate minerals, we expect that the concentration of evaporite ions should be consistent with changes in the sediment concentration and the concentration of silicate ions.*"

Lines 64-65 – Here you say that you expect that Na enhanced physical erosion will enhance chemical weathering in the dry season by exposure of fresh silicate minerals. Wouldn't you expect new evaporite minerals to be re-precipitated in the dry season at the surface? I understand that enhanced physical erosion in the wet season could expose fresh minerals in the dry season, but I suspect water throughput, not mineral availability is limiting in the dry season.

Response: We agree with this reviewer on this important point. During the dry season (November to April), we expect that the weathering rate of the mudstone bedrock will depend on humidity, rather than mineral availability, and particularly on the number of wet-dry cycles. Approximately 8.7% of annual rainfall (1983 mm) occurs during the dry season, according to rainfall data over the past 70 years (1951-2021) at Chi-ding

station (5 km from Guting Bridge, upstream sampling site). Thus, the amount of rainfall during the dry season is nevertheless substantial. During the dry season, Taiwanese mudstone experiences occasional rainfall and high air temperature, which cause the mudstone surface to form mud cracks. Mud cracks become channels for the evaporite redeposition, and channels for the infiltration of surface runoff which leads to the properties of the mudstone, e.g., rock density, water permeability, and ion concentration between the surface (a few centimeters to 10 cm depth) and bedrock are different (Fig. S1). For example, the bedrock hardly participates in physical erosion during a rainfall event due to low permeability. Therefore, bedrock weathering is the key to maintaining high erosion and high weathering in the mudstone area.

We revised the sentence as follows:
*"We interpret our observations in the badlands to reflect how the excess sodium that re-precipitates at surface in dry season enhances physical erosion and chemical weathering in the following typhoon event."*

[Figure]

Figure S1. (a) Aerial view of mudstone badlands landscape. Badlands cover an area of 4.37 km2, which accounts for 2.49% of the total catchment area. (b) the profile of a mudstone hillslope. Mud cracks extend down to the area approximately 10 cm below the mudstone surface.

Lines 169, 181, 189 – I think the endmember values would be more helpful in a table.
Response: Please see Table 1.

**Table 1** Input end-members for the mixing model.

| End-member | SO$_4$/Cl | Na/ Cl | K/ Cl | Mg/Cl | Ca/Cl |
|---|---|---|---|---|---|
| Precipitation | 0.35 | 0.90 | 0.09 | 0.18 | 0.35 |
| Evaporites | 0.6±0.6 | 1.0±0 | 0.026 | 0.1±0.08 | 0.5±0.5 |
| | Ca/Na | Mg/Na | | | |
| Silicates | 0.35±0.25 | 0.24±0.2 | | | |
| Carbonates | 60±30 | 30±15 | | | |

Line 164 – More details on your use of MEADIR would be helpful. After reading through the methods, it looks like you are only using MEADIR to apportion Ca and Mg? It is a little confusing because you assign all of the Na that is not rain or evaporite to silicates, but then give a Ca/Na ratio for carbonates. It seems slightly contradictory, but maybe the math works out such that the Na from carbonates is negligible. Nonetheless, more explanation regarding the mixing model would assuage my worries.

Response: We use MEADIR to distinguish the proportion of major elements in rainwater, evaporite, silicate and carbonate, so it is not limited to Na and Mg.

We clarify this point in the revised sentence:

*"We calculated the proportions of ion contribution **from rainwater, evaporite, silicate and carbonate for Ca, Mg, Na, Cl, and SO$_4$** with the MEANDIR inversion model (Kemeny and Torres, 2021), a MATLAB script for inverting fractional contributions of end-members, and for constraining the chemical compositions of those end-members **with Monte Carlo propagation of uncertainty**."*

Lines 155, 205- You wrote that SAR > 13 causes soil particles to repel each other, preventing the formation of soil aggregates. In the results and Fig. 2, SAR rarely goes above this threshold. Does this pose an issue for the interpretations that evaporite weathering enhances physical denudation through disaggregation/deflocculation?

Response: We agree with this reviewer on this important point. Sodium adsorption ratio (SAR) allows assessment of the state of flocculation or dispersion of soil aggregates for pore water. Calculated with pore water chemistry in the dry season from the same study site, the SAR is 240.8 and exceeds the threshold value of 13. So, we expect that soil deflocculation should exist on the hillslopes. According to Ayers and Westcot (1985), the value for irrigation water smaller than 3 is low, from 3 to 9 is medium and above 9 is high. Overall, SAR in the river water has a maximun value of 4.41 at Guting Bridge (3.14 at Nanxiong Bridge), suggesting soil deflocculation within river water is weaker than on the hillslopes. However, if we accept river water as a carrier of dissolved ions from hillslope, then the trend of river water SAR is able to reflect the extent of dissolved Na$^+$ from hillslope. We clarify this point and correct the relevant description throughout the main text.

Paragraph starting at Line 234- This section is talking about enrichment and dilution of different sources with time. I think this section would benefit of thinking about these enrichments and dilutions as a function of discharge. There are many papers that have looked at concentration-discharge behaviors during flood pulses. Discussion of how

these findings compare to others would strengthen this analysis.

Response: We agree with this reviewer on this important point. We do not have discharge data at the upstream sampling site. So, we use enriched ratios to represent the dynamic change in the chemical composition of river water during a typhoon event. Enriched ratios can directly show the difference between ion concentration at a certain time and the ion concentration at the first observation. Hence, dilution behavior or mobilization behavior was relative to the state before the event. We add relevant discussion as follows:

*"We also use the concentration–discharge (cQ) relationship of each ion at rising and recession limb, as well as baseflow at Nanxiong Bridge to assess the state of dilution behavior (Fig. S2). Overall, our results show that all ions are in a dilution, and the dilution in recession limb is stronger than that in rising limb, except for $SO_4$ during baseflow ($\theta=0.07$). The concentration of Na, Cl and K during baseflow have a higher variability than the values during the event. Additionally, Na, Cl, and $SO_4$ increase the concentration with increasing flow at the certain period of rising limb."*

[Figure]

Fig. S2. The cQ relationships at rising and recession limb of the typhoon event, and baseflow at Nanxiong Bridge. Black, blue and red colors denote the sample from baseflow, rising and recession limb, respectively. Green dashed circles denote the sample of Na, Cl, and $SO_4$ at the certain period of rising limb. Dots, lines and numbers denote samples, trendline and slope respectively.

Knapp, J. L., von Freyberg, J., Studer, B., Kiewiet, L., & Kirchner, J. W. (2020). Concentration–discharge relationships vary among hydrological events, reflecting

differences in event characteristics. Hydrology and Earth System Sciences, 24(5), 2561-2576.

Moatar, F., Abbott, B. W., Minaudo, C., Curie, F., & Pinay, G. (2017). Elemental properties, hydrology, and biology interact to shape concentration‐discharge curves for carbon, nutrients, sediment, and major ions. Water Resources Research, 53(2), 1270-1287.

Line 262- the title of this section is evaporite dissolution over time, but you discuss carbonate and silicate weathering here as well. Consider re-naming this section to more accurately reflect its contents.

Response: We agree with this reviewer on this important point.

We clarify this point in the revised title:
"*4.2 Evaporite, **silicate and carbonate** dissolution over time*"

Lines 285- 287 - I thought that you already determined the mixing by MEANDIR. What is gained by also looking at mixing with Sr?

Response: We demonstrate dynamic changes of weathering by using Molar ratio mixing, i.e., the signal shifts between silicates, carbonates, and evaporites, and identify potential contributing sources of ions by using the distance between samples and endmembers. Using ion concentration only is unable to distinguish changes in concentration due to dilution by rain or contributions from other endmembers. Sr has been used in the previous study (Gaillardet et al. 1999) which has better performance in the classification of three weathering types than Si, Mg, and K.

Fig. 5- I find this figure difficult to follow and it is not totally clear to me what new information the figure is conveying that is not already shown in Fig. 3a-c.

Response: Figure 5 has a better performance in showing dynamic paths of weathering during the rainfall event, the difference from the based period, distance from the state before the event, and distance from the endmembers. We also adjusted Figure 5 to make it easier to read.

Line 338- Is this true? I thought that evaporite-derived solute fluxes were larger than the silicate-derived fluxes.

Response: During the typhoon event, evaporite-derived solute flux is 10.9 ton/km$^2$/yr which is less than silicate weathering.

We clarify this point in the revised sentence:

*"Our observations show that silicate weathering during the typhoon event contribute 16.8 ton/km²/yr, contributing 16.6% to the annual silicate weathering flux (Table S3)."*

Lines 346-348- Your mixing calculations should tell you the proportion of Ca and Mg from carbonate and silicate weathering. It is unclear to me how the enriched ratio of carbonate Ca means that it is not a contributor to weathering fluxes.

Response: Here we discuss the ratios in figure 5 that approach the silicates/carbonates weathering (high Na/Sr, high Ca/Sr ratios) after the peak discharge. We identify dominant weathering by filtering out major ion contributors. After the peak, Na, Ca in silicates and evaporites are greater than 1, whereas the Na, Ca in carbonates are continuously diluted. However, dilution does not mean that Ca is not contributing to weathering fluxes. It only indicates that $Ca_{carb}$ contributions are greater during the non-typhoon season.

We clarify this point in the revised sentence:

*"…, but the increased $Na^+$ and consistently enriched ratio of carbonate $Ca^{2+}$ does not make this a likely **main** contributor to the Erren River **during the typhoon event**."*

Line 357 – I am a little lost here. Are you suggesting that increased erosion on the hillslope increases silicate weathering in the river channel via abrasion of sediments in the river channel?

Response: As mentioned above, evaporite ions ($Na^+$) dissolved by rainwater enhance hillslope erosion and cause high concentrations of suspended load and weathered silicate minerals, and substantial suspended matter provides material for abrasion in the channel. Masses of Na and Ca are reduced by 10.6% and 9.9%, respectively, in the suspended sediment during the course of the typhoon event, which we interpret as abrasion in the channel may be responsible for it. The study of catastrophic glacial lake outburst flood in Nepal documents that mechano-chemical dissolution of weakly bound ions, e.g., F- from the fresh muscovite surfaces is driven by abrasion under the high energy sediment transport with reorganization of the river bed (Andermann et al., 2022, https://doi.org/10.5194/egusphere-egu22-10417).

Line 365 – Your flux is comparable in what way? Same magnitude?

Response: We clarify this point in the revised sentence:

*"The global annual silicate weathering flux of rivers is 15.7 ton/km²/yr (Gaillardet et al. 1999), relative to our value of 16.8 ton/km²/yr."*

Lines 369-370 – I don't think this is true. In my experience, most global weathering calculations account for evaporite weathering.

Response: We revised the sentence as follows:

*"Evaporites, including halite (NaCl) and gypsum (CaSO4), are found in few sedimentary environments, and they are often excluded from **the estimation of $CO_2$ consumption** (Gaillardet et al., 1999)."*

Lines 411-412- Again, I do not think your data supports this. Table S3 shows that at baseflow, carbonate-, silicate-, and evaporite-derived fluxes are roughly equal and during the typhoon, silicate- and evaporite-derived fluxes are roughly equal.

Response: For the non-typhoon days, total weathering is 344.1 ton/km$^2$/yr. So, relative to $TDS_{sil}$ (29%) and $TDS_{evap}$ (28%), the greatest single contributor to total weathering comes from carbonates (39%). In contrast, we observe the opposite trend during the typhoon event, where the $TDS_{carb}$ contributes the least. In contrast, $TDS_{sil}$ and $TDS_{evap}$ are similar.

We revised the sentence as follows:

*"Our observations show that the water chemistry of the typhoon event is **mainly contributed** by silicate weathering at 16.8 ton/km$^2$/yr **and evaporite weathering at 10.9 ton/km$^2$/yr**, in contrast with baseflow (non-typhoon) conditions that are **mainly contributed** by carbonate weathering."*
* * *
Line 9- "manifestations" is an awkward word choice, consider rewording

Response: We revised the sentence as follows:

*"Chemical weathering driven by physical erosion is a natural process that strongly affects chemical and solid matter budgets at the Earth's surface."*

Line 49- "SW" needs to be defined. Intuitively this means "southwest", but its better to define here and abbreviate after.

Response: We revised the sentence as follows:

*"Previous studies in the badlands of **southwestern (SW)** Taiwan have revealed that dissolving halite and gypsum at depth migrate to the slope surface and deposit in desiccation cracks during the dry season (Higuchi et al., 2013, 2015; Nakata and Chigira, 2009)."*

Line 62- I would add the total length of time here. "… a temporal resolution of 3-hours

collected over XX hours."

Response: We revised the sentence as follows:

*"…, collected **at** a temporal resolution of 3 hours **over 3 days.**"*

Line 207- It might be good to define D50 in the methods for grain size.

Response: We revised the sentence as follows:

*"**The median grain size** ($D_{50}$) ranged from 3.9 to 8.2 μm, with an average value of 5.6 μm during the second pulse, …."*

Line 275- are the values in parentheses correlation coefficients?

Response: We revised the sentence as follows:

*"At Guting Bridge, all evaporite ions have a statistically significant positive correlation with the corresponding silicate ions ($Na^+$, $\rho$ = 0.98; $Ca^+$, $\rho$ = 0.81; $SO_4^{2-}$, $\rho$ = 0.98, p<0.05)."*

Referee #2

1. In this calculation of ions' source, the global endmembers of Gaillardet et al., (1999) were used. Are these values suitable for this study? The results show that the silicates increase during the rainstorm. Why does the silicate-sourced ions (e.g., Na) increase with high runoff? Most studies have shown that silicate-sourced ions decrease with high runoff, because of the slow weathering kinetics. I think the calculation was highly depended on the values of endmembers.

Response: In the absence of better local constraints, we select published global endmembers and a few regional studies using the same endmembers set for local constraints, e.g., Bufe et al (2021. https://doi.org/10.1038/s41561-021-00714-3). We agree that the use of global endmembers leads to a larger range of estimations, but is still appropriate in discussing trends in weathering rates. Silicate-sourced ions proportionally increasing with peak discharge can be attributed to two mechanisms: 1. weathered silicate minerals abrasion in the channel, see section 5.2; 2. upstream sampling site is adjacent to the solute source, cQ relationship is transport-limited rather than kinetic-limited.

2. For the SO4, how does it from silicate and carbonate? Does silicate and carbonate contain SO4?

Response: Silicate and carbonate liberate $SO_4$, but traditionally in calculating the weathering rate, $SO_4$ after deducting the contribution of rainfall and evaporite should be classified as sulfate weathering and human pollution. Within the calculation of MEADIR, we set $SO_4$ from silicate and carbonate as 0 to achieve it. Therefore, we agree

with the reviewer to delete the contribution from silicate and carbonate of SO$_4$ in Figure 4 and related descriptions in section 4.2. However, this removal will not affect the calculation of weathering flux on silicates and carbonates, because the ion concentrations of SO$_4$ from silicates and carbonates are negligibly low (~$10^{-4}$ $\mu$ mol/L).

3. What is D50? It's explaination should be shown in the maintext

Response: We revised the sentence in line 226 that appeared for the first time as follows: *"**The median grain size** (D$_{50}$) ranged from 3.9 to 8.2 μm, with an average value of 5.6 μm during the second pulse, …."*

4. For the enriched ratio, the authors can show using an equation. In addition, the authors used the first observation. The average value is much better. It is best use the annual average, at least the average during the sampling time.

Response: We added the clarification of calculation of the enriched ratio. Please see Line 157. In this study, we are interested in understanding how the river chemistry changes from the initial, background values, so we have decided to keep our approach.

5. For the evaporite endmember, the ratios are affected by the mixing of gypsum and halite. Can this value express the evaporites in this study? Chao et al., 2013 can not be found in the references.

Response: In MEADIR, we set Ca/Cl as 0.5±0.50 which means the concentration of gypsum to halite is between 0-1. Based on our measurement, the concentration of Ca is roughly half the concentration of Cl, baseflow is 0.51±0.18 and the typhoon period is 0.50±0.14 which falls within the range of setting. Therefore, we accept the endmember of evaporites.

Thanks to the reviewer for pointing out the typo, we correct the citation to Chao et al., 2011.

Chao, H.-C., You, C.-F., Wang, B.-S., Chung, C.-H., Huang, K.-F.: Boron isotopic composition of mud volcano fluids: Implications for fluid migration in shallow subduction zones. Earth and Planetary Science Letters, 305. https://doi.org/10.1016/j.epsl.2011.02.033, 2011.

6. Did the authors consider the carbonate precipition? The evaporite dissolution express Ca, which can enhance carbonate precipitation.

Response: We are not quite clear on the question, but we did not consider or calculate how evaporite dissolution might result in secondary carbonate precipitation. So, we calculate the saturation index of our samples using PHREEQC (Parkhurst and Appelo, 1999) The result shows that calcite and dolomite are unsaturated (saturation index <0) most of the time, and therefore the impact of carbonate precipitation on carbonate weathering flux could be negligible (Figure 1). Moreover, when Ca and Mg precipitate in the form of hydroxide or carbonate with increasing pH value (Pourboix, 1974), then Na would gather near the clay flakes, which increases osmotic pressure and water content, resulting in softening soil."

Parkhurst, D. L. and Appelo, C. 1999, User's Guide to PHREEQC (Version 2): A Computer Program for Speciation, Batch-Reaction, One-Dimensional Transport, and Inverse Geochemical Calculations Water-Resources Investigations Report 99-4259, US Geological Survey.

Pourboix, M.,1974. Altas of Electrochemical Equilibrium in Aqueous Solutions" National Association of Engineers, Waston Texas, U.S.A.

[Figure]

Figure 1. Timeseries runoff (Q), saturation index of calcite and dolomite at Nanxiong Bridge. The blue line denotes hourly discharge. The red line denotes the saturation index of calcite, and the black line denotes the saturation index of dolomite. The pink area indicates the saturation index below 0.

7. Line 338-339, the authors should show what proportion of the discharge in this study occupying in the year. Not just the silicate weathering
Response: In section 5.2, we discuss that the silicate weathering flux during the event

in this study site is comparable to the global annual silicate weathering flux of rivers. Therefore, we only show silicate weathering flux.

---

## Author Response (AR2)

**Response Letter**

We thank the reviewer for the insightful and constructive comments and corrections, which helped us to greatly improve the manuscript. We address their concerns point by point, and highlight implemented changes in the manuscript.

Comments with line number on the revised manuscript:

12 – remove ", homogenous" - I'm not sure that is true at any scale!

Response: We revised the sentence as follows:

*"Badland landscapes formed in highly erodible substrates have the potential to respond to individual events on scales that are rapid enough for direct observation."*

17 – this sentence isn't clear. Which dominates the weathering signal? Rephrase the second part.

Response: We removed the redundant sentence.

*"Evaporite weathering at peak rainfall is succeeded by peak silicate weathering at maximum discharge."*

Equation 9 – "carb" instead of "crab"

Response: We corrected the typo.

207 – specify that the model used via Meandir

Response: We revised the sentence as follows:

*"Table 1 Input end-members for the MEANDIR inversion model."*

367 – reporting the flux during the typhoon is not the same as the typhoon contributes to the silicate weathering flux – see discussion in Calmels et al., 2011, Earth & Planetary Science Letters. I would rephrase – the typhoon provides the mechanism to mobilise weathering products from the catchment. The timescale of precipitation to water flow is much too short to have silicate mineral dissolution, and so it is important to consider instead the role the typhoon plays in the hydrological mobilization of weathering products.

Response: We agree with this reviewer on this important point. So, we revised the sentence as follows:

*"Our results show that the typhoon is responsible for mobilizing 16.8 ton/km$^2$/yr of dissolved solutes derived from silicate weathering during the course of the event, and this flux corresponds to 16.6% of the annual silicate weathering flux (Table S3)."*

380 – again this is potentially misleading – silicate weathering at these magnitudes and rates cannot be happening during the typhoon which lasts hours? – flood events mobilise older water from catchments and move it to the outlet (e.g. Calmels et al., 2011), meaning a large proportion of solutes are not produced during the event

Response: We agree with this reviewer on this important point.

So, we revised the sentence as follows:

*" We also observe a 10–18% loss in the individual concentrations of Ca, Na, Al, and Sr in the suspended sediment during the course of the typhoon event, whereas concentrations of Fe, K, Mg, and Mn increase by 3-10% (Table S6). The dissolution kinetics of silicate weathering are multiple orders of magnitude slower than carbonate or evaporite weathering (Meybeck, 1987), suggesting that significant weathering of fresh silicate minerals over the course of a single typhoon event is unlikely. Thus, the observed changes in ion concentrations during the event are likely to arise from heterogeneities in the bedrock composition or the input of previously weathered silicate minerals from a deeper groundwater reservoir (Calmels et al., 2011), which is different from groundwater source of baseflow during non-typhoon period. However, quantifying the role of a deeper groundwater inputs is difficult in the absence of isotope data."*

381 – you cannot use sediment concentration data the way shown here to look at "loss" during weathering. See comment on Table S6 below. Please update appropriately.

Response: We agree with this reviewer on this important point.

So, we added other elements to table S6 and recalculate total loss of mobile element by Anderson et al (2002).

394 – what is the reduced mass ?– is this relating to Table S6 – if so see comment below.

Response: we revised the sentence as follows:

*"We suggest that high suspended sediment concentrations, combined with high energy flow during the typhoon, caused increased silicate input from the weathered silicates in the suspended sediment, which has also been observed in typhoon-driven silicate chemical weathering from silicate minerals at surface (Meyer et. al., 2017)."*

417 – what about the mechanical properties of these rocks and the fact that typhoons deliver very intense precipitation – I think these would overshadow these chemical effects?

Response: We agree that physical erosion caused by typhoon rainfall may exceed chemical erosion. Our focus is not to compare the magnitude of the two types of

erosions. Rather, we highlight the significance of chemical erosion resulting from the dissolution of evaporites, which is often overlooked in mudstone regions. Several studies have demonstrated that the presence and distribution of evaporites on slopes can influence the development of gullies.

Figure S1 – please give approximate location of the images taken with reference to the map figure 1.

Response: We added the geographic reference of Figure S1 into the caption.

"Fig. S1. (a) Aerial view of mudstone badlands landscape taken from Guting (GT) Bridge (Figure 1)…."

Table S6 – Why is only Ca and Na concentrations on the solids provided? It would be useful to provide an overview of the sediment chemistry if it was measured with standards and quality control, as it is suggested it was measured by OES.

Response: We added other elements to table S6 as follows:

Table S6 the sediment chemistry of Guting Bridge.

| duration | Ca | Na | Al | Fe | K | Mg | Mn | Ti | Sr |
|---|---|---|---|---|---|---|---|---|---|
| hr | | | | | µg/g | | | | |
| 5.5 | 10914 | 9981 | 78214 | 42084 | 23413 | 11294 | 524 | 4901 | 101 |
| 41.5 | 9833 | 8919 | 86454 | 48893 | 28034 | 13157 | 552 | 5319 | 99 |
| Total loss, $\tau$ (%) | -17 | -18 | -17 | 7 | 10 | 7 | 3 | - | -10 |

Second, and more importantly, this loss ratio is potentially misleading. Changes in solid concentration of an element could reflect relative loss of gain (as interpreted here). But it could also reflect i) heterogeneities in the bedrock composition, which could easily by >10%; ii) changes in the relative role of mineral grains – e.g. both Ca and Na could decrease while Al or Si increase, as the proportion of different dominant phases shifts. Therefore, with concentration alone, you cannot report a "loss" from the sediments. To do that, studies will often normalize the mobile element (Ca or Na) to an immobile element (e.g. Zr, Ti, or Al). See work by Brantley and co-workers at the Shale Hills CZO for a number of examples, and then please re-phrase.

We agree with this reviewer on this important point. We recalculate total loss of mobile element by Anderson et al (2002) which method was applied to Kim et al (2018). Notably, we do not use the parent rock as a reference when calculating the total loss, but the suspended sediment before the typhoon event as the reference. Given that the change in total loss less than ±20%, we agree that the differences should be due to parent rock heterogeneity or dissolution of weathered material. So, we revised the relevant sentence based on the statement.

---

## Author Response (AR3)

**Response Letter**

We have checked the complete manuscript, as well as the misspellings. We also updated the figure 1a in the supplementary for better resolution.